# Logical Tasks for Measuring Extrapolation and Rule Comprehension

## Abstract

Logical reasoning is essential in a variety of human activities. A representative example of a logical task is mathematics. Recent large-scale models trained on large datasets have been successful in various fields, but their reasoning ability in arithmetic tasks is limited, which we reproduce experimentally. Here, we recast this limitation as not unique to mathematics but common to tasks that require logical operations. We then propose a new set of tasks, termed logical tasks, which will be the next challenge to address. This higher point of view helps the development of inductive biases that have broad impact beyond the solution of individual tasks. We define and characterize logical tasks and discuss system requirements for their solution. Furthermore, we discuss the relevance of logical tasks to concepts such as extrapolation, explainability, and inductive bias. Finally, we provide directions for solving logical tasks.

## 1 Introduction

Human production activities rely on logical operations and unconscious, automatic, or intuitive inference. The behavior of existing neural networks is closer to intuitive inference. These networks do not have mechanisms suitable for utilizing seen data repeatedly, nor the ability to explicitly and efficiently use patterns and rules hidden in the dataset. Dealing with these instead requires logical operations. We considered that developing novel architectures which deal with such logical operations was an attractive research topic.

Large neural networks trained on significant amounts of data have achieved great success in many fields, such as image generation, text generation, and more. GPT (Alec Radford & Sutskever, 2018) and BERT (Devlin et al., 2019) variations in natural language processing are intensively studied. A state-of-the-art model, PaLM (Chowdhery et al., 2022), shows tremendous ability to provide plausible outputs in reasoning tasks, such as explaining jokes and chaining inferences. However, even though PaLM has 540B parameters and was trained on 780 billion tokens, its accuracy in mathematical reasoning, which requires strict and logical multi-step operations, is limited (Chowdhery et al., 2022). In addition, the data efficiency of these large neural networks approaches is poor. A couple of million examples are required to acquire the ability to add and subtract. Moreover, even after training on a few million examples, neural networks do not work well in the extrapolation regime. These facts indicate that an approach based merely on increasing both model parameters and training data is not enough.

Overcoming this situation requires the development of novel architectures or inductive biases. However, inventing these is no easy task. As a prior step, in this paper we review the current situation, clarify the problem, and indicate directions. Our look into the limitations of current deep learning technologies and design of new architectures will be aided by a focus on extrapolation behavior in elementary arithmetic tasks. To succeed in extrapolation, systems must capture and use rules hidden in the dataset in either an explicit or implicit way, enabling coherent predictions.

It is not our goal to develop a model that merely solves mathematics in any ad hoc way. Instead, we would like to develop a system that can extrapolate by learning from data. To this end, it is crucial to focus on the more general, logical tasks proposed in this paper and develop neural networks that can solve them. It is essential to refrain from introducing task-specific domain knowledge, and from developing specific advanced

skills. To achieve this purpose, this paper collects datasets proposed in different contexts but which share common properties in terms of logical tasks.

After reviewing related work in section 2, we propose logical tasks and give their properties, system requirements, and examples in section 3. In section 4, we review and introduce arithmetic tasks as an example of logical tasks, and in particular deal with the addition task to investigate the limitations of neural networks. Our implementation of this experiment is available.[1] In section 5, we discuss how logical tasks relate to extrapolation, explainability and the possible novel inductive biases for logical tasks. Finally, we provide a conclusion in section 6. We also give details of the experimental setup in appendix A, and the result of a supplementary experiment in appendix B.

Our contributions are that:

- we propose logical tasks, namely a new set of tasks which will be important to overcoming the shortcomings of current AI systems.

- we clarify the relationships between logical tasks and some concepts, such as extrapolation, inductive biases, and interpretability.

- we introduce a simple experimental setup to study the extrapolation behavior of neural networks, and reveal the practical shortcoming that neural networks cannot extrapolate well even in the addition task, which is the simplest logical task.

- we give future directions on how we will create novel systems solving logical tasks.

## 2 Related Work

### 2.1 Mathematics Dataset

**Datasets** Mathematics requires logic, which makes it difficult for neural networks to solve. Mathematics Dataset (Saxton et al., 2019) is the standard dataset in this area. It consists of a few dozen modules, such as algebra, arithmetic, comparison, polynomials, etc, and provides an implementation of generators of modules as well as a pre-generated dataset. This dataset is frequently used to develop novel architectures and examine the behavior of neural networks.

GSM8K (Cobbe et al., 2021) is a dataset with a similar level of difficulty to Mathematics Dataset. A feature is that the questions consist of several sentences written in natural language rather than pure mathematical expression. Both datasets are limited to school math. Another direction is to include higher-level mathematics. Hendrycks et al. (2021) proposes MATH, which provides more complicated tasks such as integral and differential equations. Megill & Wheeler (2019) provides a dataset for theorem proving.

All of these tasks can be considered part of the logical task. These papers often aim at solving specific tasks themselves, which frequently results in their choosing difficult mathematical problems. In contrast, our motivation differs from these papers in that we do not aim to solve specific tasks or upgrade tasks, but rather to capture common properties included in such tasks. Our goal is to develop inductive biases that are effective for a wide range of tasks, and not limited to mathematics.

**Models** Elementary arithmetic problems are expected to be solved by multi-step operations. Given an environment for a system with defining states and actions, the formulation of the problem can be seen as a reinforcement learning framework. Chen et al. (2018) presents the Neural Arithmetic Expression Calculator in hierarchical reinforcement learning and curriculum learning. Nye et al. (2022) introduces Scratchpad, a supervised learning setup with the computational processes as labels and which successfully predicts a certain extrapolation regime. Some studies (Wei et al., 2022b), (Nogueira et al., 2021), (Cognolato & Testolin, 2022) (Hu & Yu, 2020) attempt to solve mathematics by introducing modules tailored to the characteristics of the task.

---

[1] The URL will be added.

## 2.2 Large Language Models

GPT-3 (Brown et al., 2020a) is a typical large language model (LLM) for generating texts. Recent models such as PaLM work much better, however. In particular, PaLM gives very plausible outputs for explaining jokes and inference chaining (Chowdhery et al., 2022). The common principle of these models is that they have many parameters and are trained on a huge amount of data. This trend of increasing model and data size has continued for the last few years. It is a promising approach for future intelligent systems that we have yet to encounter. This trend does not end only in natural language processing, however. In image generation, for example, DALL-E (Ramesh et al., 2021) (Ramesh et al., 2022) shows an astonishing ability to generate images that must be unseen in the training dataset, while in program synthesis, Codex (Chen et al., 2021) succeeds in generating codes from natural language at a certain level to aid software engineers.

Nevertheless, these models are poor at generating data involving the performance of rigorous operations. For instance, they face challenges in specifying subtleties likely desired in future applications, such as the structure of paragraphs or images (Marcus et al., 2022), a brush touch, etc.

Mathematics requires strict multi-step operations and is therefore an archetypal task for demonstrating the shortcomings of LLMs. There are no reliable large language models to do this with symbolic operations. GPT-3 fails to solve elementary arithmetic equations. It works well on two-digit addition, but accuracy decreases as the number of digits increases (Brown et al., 2020b). The accuracy of PaLM on GSM8K is also limited, at only 58% (Chowdhery et al., 2022). Minerva (Lewkowycz et al., 2022), which is PaLM trained on additional data related to math, outperforms the original PaLM, but its accuracy remains at 50.3% on MATH (Hendrycks et al., 2021) and 78.5% on GSM8K.

## 2.3 Probes of Neural Networks

Despite increasing clarification that neural network capability improves as the size of the model increases, the limitations and nature of these networks remain unclear. Illustrating this, BIG-bench (Srivastava et al., 2022), a benchmark of tasks that are not solvable by LLMs, includes over 200 tasks from fields as diverse as linguistics, mathematics, biology, software development and so on, and thereby reveals the properties and limitations of LLMs.

Nogueira et al. (2021) investigated the limitations of Transformers on addition tasks by changing the model size and representation of numbers. The authors found that the introduction of position tokens enabled highly accurate predictions. Wei et al. (2022a) reviewed how abilities for various tasks emerge in LLMs.

Power et al. (2022) reported a new phenomenon, *grokking*, characterized as delayed generalization far after overfitting on small algorithmic tasks. Liu et al. (2022) further studied this phenomenon by focusing on addition and permutation as toy models. They defined four learning phases and drew phase diagrams. Since this phenomenon was observed when both the dataset and model size were small, the experimental setup is more straightforward than that of LLMs, and will enable the theoretical study of generalization.

There are divergent directions to studying the behavior of neural networks. Our experiment will shed light on this behavior by following another new direction, based on the notion that the simple setup of the addition task is suitable for grasping how the extrapolation ability occurs, as we show in section 4.3.

## 2.4 Neuro Symbolic AI

Neuro symbolic (NeSy) AI is a subfield of AI that pursues the combination of neural networks and symbolic AI systems. On the one hand, neural networks are composed of many trainable parameters in which information about tasks is kept in a difficult way for humans to understand. They function well for complex high-dimensional data when massive data are available, but they have the disadvantages of black-box behavior, difficulty making stable predictions, and low explainability. On the other hand, symbolic AI, which aims to develop systems dealing with symbols, has strengths and weaknesses that contrast with neural networks, such as higher explainability but difficulty fitting complex data. Symbolic AI includes systems that handle first-order predicate logic, knowledge graphs, and natural language processing.

NeSy AI aims to mitigate these weaknesses and reinforce their strengths by integrating the two contrastive systems. Specifically, we expect that NeSy AI systems will have functions such as out-of-distribution (OOD) generalization, interpretability, learning from small data, error recovery, transferability, and reasoning (Sarker et al., 2022) (Hamilton et al., 2022).

Hamilton et al. (2022) categorizes studies in NeSy AI using Kautz's categories (Kautz, 2020). Their paper points out that the evaluation criteria in these studies are divergent because of the absence of a common dataset. Since it is difficult to compare these studies, one of the main issues in the field is the establishment of standard benchmarks.

The purposes of our proposed logical tasks overlap those of NeSy AI. The review papers above (Sarker et al., 2022) (Hamilton et al., 2022) provide a survey and review of studies and categorize them to clarify their positions. In contrast, our paper reviews more technical aspects, such as extrapolation and inductive biases, and mentions the relationships between them and logical tasks. In addition, our novel perspective of logical tasks indicates a direction to compensate for the shortage of benchmarks in NeSy AI.

### 2.5 Inductive Bias

Inductive bias (Mitchell, 1980) (Gordon & Desjardins, 1995) is a bias built into a model to maintain appropriate predictions on test data, namely generalization, based on the idea that a model's behavior is limited when it is trained on finite data only. In other words, the developer's knowledge is incorporated into the model in advance. For example, the convolutional neural network is an inductive bias adequate for image recognition tasks, while Transformer (Vaswani et al., 2017), which has an attention mechanism (Bahdanau et al., 2015), is considered adequate for serial data (Goyal & Bengio, 2020).

In general, there is a trade-off between domain specialization and algorithmic generality. Domain-specific inductive bias improves accuracy and makes learning easier for a specific task. However, it is also likely to make it ineffective for other tasks due to its reduced generality. On the other hand, complex tasks can not be solved without any inductive bias.

When the developer's knowledge is appropriately introduced, it improves the accuracy of concerning tasks and facilitates optimization in challenging tasks. However, this introduction of knowledge can also be regarded as humans partially solving the task, which in turn prevents models from learning from the data. As Chollet (2019) discusses, task-specific knowledge induced by developers should be a matter of concern.

Introducing a moderate inductive bias is suitable for many tasks, whereas an excessive bias is only for specific tasks. Development of extreme biases results in tasks being solved by developers rather than by the systems themselves. The degree of developer knowledge corresponds to the degree of domain specificity: the more task-specific the bias, the less versatile the system tends to be.

In this paper, we ask what inductive biases are suitable for logical tasks. To this end, we clarify what the logical tasks are.

### 2.6 Extrapolation

Extrapolation, the antonym of interpolation, refers to the prediction of data which differ from the training dataset with regard to region or nature. This terminology is widely used in different machine learning fields, such as curve fitting, out-of-distribution (OOD) generalization (Shen et al., 2021), and arithmetic tasks. Since the main scope of machine learning is to develop systems that can work in new situations, extrapolation is an important concept. However, a precise mathematical definition of this word is difficult and its meaning remains ambiguous.

The simplest usage is in curve fitting. Considering one dimensional space, we can easily define the section wherein data points exist $[x_{min}, x_{max}]$. Prediction in this regime is called interpolation, and otherwise it is called extrapolation. Although curve fitting is the simplest case, there are other nontrivial natural definitions. If some clusters are composed of data points, blanks between them can be seen as the extrapolation regime (Ebert et al., 2014). Moreover, we can also consider more complex regions, such as a rectangle hull, convex hull, and concave hull, as high dimensional cases.

Table 1: **The summary of the definition of logical tasks and the properties of systems.**

|  | **Definition of logical tasks** | **System requirements for logical tasks** |
|---|---|---|
| **Symbols** | The dataset is represented by symbols, which are discrete representations. Combinations of symbols have meanings, and are task-specific. | Systems have to deal with symbols corresponding to inputs and outputs, as well as to those invented by themselves. |
| **Rules** | Rules of the task determine the relationships among symbols. There is a finite number of rules in logical tasks. Ideally, all examples obey the rules without exception. | Systems must capture the rules and change them into operators to use in the algorithm. |
| **Algorithms** | There exists an algorithm to solve the logical task when the rules are known. | Systems must find an algorithm to solve all examples and follow it rigorously. To solve with algorithms means that the process requires multiple steps. |

Since OOD generalization, from its definition, argues for prediction on data not seen in the training dataset, it can also be regarded as a kind of extrapolation. High-dimensional data such as images, audio, and natural language are often used in this context. Regarding images as data points may allow us to introduce consecutive space for images based on the manifold hypothesis (Cayton, 2005). However, this is much more ambiguous than curve fitting, because an appropriate definition of the distance between images is nontrivial, and highly dependent on the task under consideration. Balestriero et al. (2021) defines interpolation as a sample whenever it belongs to the convex hull of a set of samples; otherwise, it is extrapolation.

In arithmetic tasks, there are only dozens of discrete symbols. A data point is composed of these several symbols. This setup is the same as natural language processing. Given $N$-digit integers as training data, the extrapolation regime is $(N + 1)$-digits or larger integers.

## 3 Logical Tasks

We propose the new concept of logical tasks, which we define as tasks that require logical operations, such as mathematics. The *logical task* as generic terminology refers to a class of tasks with several distinguishing properties, as tasks such as object detection and semantic segmentation in image recognition have common properties. We group these tasks and define a new category to discover new inductive biases. We give the definition and properties of logical tasks in section 3.1 and system requirements in section 3.2; argue our purpose in section 3.3; and list specific examples in section 3.4.

### 3.1 Definition and Properties

#### 3.1.1 Definition

We define *logical tasks* as tasks in which symbols represent inputs and outputs, the symbols are subject to rules, and an algorithm utilizing the rules determines the outputs from the inputs unambiguously. Given input data, systems must return the output correctly. To specify more clearly, we explain them from three aspects: symbols, rules, and algorithms. We also summarize this in Table 1.

**Symbols** Symbols are discrete representations which represent the inputs and outputs of logical tasks. There are finite kinds of symbols in a task, but the number of these combinations can be quite significant. Examples in the dataset are composed of multiple symbols.

The meanings assigned to symbols depend on the rules of the task. The assigned meanings can differ from the usual context. This feature hinders utilization of a pre-training strategy. An example is the addition task with random permutation of Arabic numerals.

In a complex setup, data can be represented in high-dimensional vectors such as images or sound, but we should consider this factor separately from the original logical tasks. For example, we can include some fluctuations of representations, such as introducing images to represent symbols of numbers using MNIST (Lecun et al., 1998) or SVHN (Netzer et al., 2011).

**Rules**   Every logical task has its own set of rules. This set of rules determines the relationships among symbols. For instance, regarding $a+b=c$ in the addition task, $c$ is entirely determined by the left-hand side. In this way, the rules of logical tasks determine the relationships among symbols and apply to all examples.

A logical task consists of a finite number of rules. These rules are applied to symbols. As the number of kinds of symbols increases, the number of rules also increases. For example, in the addition task in $N$-base representation, the number of independent relationships among symbols increases when $N$ increases.

Logical tasks have no noise at all in the ideal situation. All examples obey a set of rules without exception. This nature allows us to solve them by simple algorithms. However, we can introduce anomalies that do not follow the rules. For example, in Power et al. (2022), the authors insert such anomalous examples by randomly changing answers in simple algorithmic tasks.

**Algorithms**   An algorithm is available that unambiguously determines the outputs from inputs when the rules are known. It is not trivial to induce the algorithm by reflecting the rules. The solution algorithm of logical tasks is one realization of exploration which is aimed at deriving the outputs.

### 3.1.2   Properties

**Difficulty**   Logical tasks have various difficulties. Factors related to these difficulties include the number of kinds of symbols, the number of symbols in examples, the number of rules, the description length of the rules and algorithms, and the number of steps of execution. In general, the space to be explored becomes large when the values of these factors increase. This makes logical tasks more difficult. For instance, since usual machine learning systems have to identify rules from among finite examples, increasing the number of rules makes tasks more difficult.

**Our Target**   The description of logical tasks is general, and therefore covers a range of tasks. These include NP-hard problems, which require enormous exploration, and text generation, which may have many rules that are difficult to enumerate completely. We focus on easy logical tasks from the point of view of difficulty. Examples in section 3.4 appear to be easier, but it is known that neural networks cannot solve them well. We consider that tasks requiring much exploration are composite problems that require methods to assemble and utilize knowledge of other tasks to reduce the cost of exploration, such as continual learning (Parisi et al., 2019).

**White-box Algorithms**   We introduce the concept of *white-box algorithms* from the above perspective. We define *white-box algorithms* as those algorithms that are short enough that humans can represent them in natural language or programming language and execute them in practice. In general, algorithms include anything from simple, understandable procedures to neural networks which require a massive number of execution steps and which are complicated for humans to understand. The concept of white-box algorithms excludes such complex algorithms. This definition is relative and qualitative, but it indicates our aim.

### 3.2   System Requirements

As we mention in sections 3.3 and 5, our purpose in proposing logical tasks is to develop systems with high interpretability, data efficiency, and extrapolation ability. We describe the requirements that such systems would have: stated concisely, they are required to find the rules of logical tasks, transform them into operators, identify an algorithm, and then follow it strictly. We discuss these requirements separately in terms of symbols, rules, and algorithms, and summarize them in Table 1.

**Symbols**  The systems utilize representations corresponding to the inputs and outputs within them. In addition, the systems must invent and operate novel discrete representations, such as embeddings, that work only within them.

**Rules**  Systems have to capture rules and incorporate them into the algorithm. They have to define operators; that is, functions that represent specific relationships between symbols they focus on to operate at the moment. These operators are called in a series of procedures termed an algorithm. Combinations of operators can be called skills. A skill can be considered a single-step process, but should be unfolded to fundamental operators if needed.

When humans solve logical tasks, we can use the rules described in natural language as clues. However, it is hard for machine learning models to use them. Instead, they have to learn from finite training examples. The models must explore to identify the rules from finite data, which can cause extrapolation failure. We discuss this further in section 5.2.

**Algorithms**  The systems must find an algorithm to solve a logical task and follow and execute it. Individual implementations do not clearly distinguish between the two procedures, but must perform both eventually. Executing separately and explicitly these two procedures causes robust extrapolation, since following the obtained algorithm will prevent distraction by a new combinations of symbols.

### 3.3  Purpose

The logical task will help design and develop new architectures or inductive biases as convolutional neural networks developed for image recognition and recurrent neural networks for serial data. Image recognition is a generic term in which systems deal with images as the input and/or output. In these cases, we can form a set of tasks naturally without daring to define the class since image data and audio data (as a time series) are distinctive data formats whose definition requires little or no deliberation. In contrast, the logical task is more abstract and less noticeable than these, but can be regarded as a group of tasks.

The primary purpose of proposing the logical task is to find novel architectures to solve it. However, the question is not one of simply hoping to find systems to solve a particular logical task. In other words, it is not our target to develop merely accurate models by simple reliance on domain-specific knowledge implemented as the knowledge of developers only (Chollet, 2019). In the same way that the feature engineering for solving MNIST classification is inadequate for other image recognition tasks, systems that solve only arithmetic tasks are useless. Instead, we expect to focus on developing systems that can learn from data and solve various tasks that require logical operations, rather than specific tasks only. Simultaneous consideration of multiple tasks, namely a set of tasks, such as logical tasks, mitigates over-specialization in one task.

In addition, tackling logical tasks links to improvement of the low data efficiency of neural networks. As we discuss in section 5.2, logical tasks have a close connection to extrapolation, which is finding a way to generalize in order to ensure fitting onto infinite data. The number of data points in logical tasks can be infinite, but the number of rules is finite. The rules restrict how examples are composed. Logical tasks ask whether systems can extrapolate by comprehending finite rules. Once systems capture the rules, logical tasks allow them to make almost entirely correct predictions.

We do not expect systems that are successfully able to extrapolate for any task. Indeed, at this moment, we do not know how much extrapolation is possible for any given task. Creating interpretable systems for logical tasks solved by non-white-box algorithms is challenging. For these reasons, we concentrate here on logical tasks solved by white-box algorithms. Although our scope is thus necessarily restricted, developing suitable inductive biases for logical tasks can be achieved because various tasks can still be considered.

Image recognition is not included in logical tasks because they are not expressed by symbols. However, we can for example regard objects obtained by object detection as symbols. Humans acquire generalization, interpretability, communication, and knowledge transfer by combining intuitive, complicated processes like image recognition with other processes written in white-box algorithms. An ultimate goal is the mixture of neural networks giving intuitive inferences and white-box algorithms giving logical inferences.

### 3.4 Examples

Although we regard many tasks as logical tasks, we focus here on simple ones. We have gathered datasets that appear in different contexts in machine learning but have common properties, and regard them as logical tasks.

**Addition Task**  The task is to calculate the summation of several figures represented as symbols, not numerals. In the 10-base addition task, numbers and equations are composed of a dozen symbols, namely ten independent symbols representing numbers 0-9 and + and =. We can also consider $N$-base addition. $N$ can change the number of minimal hidden independent rules in the task.

**Arithmetic Task**  Arithmetic tasks include addition, multiplication, subtraction, and division. We consider this task one of the most basic and tractable logical tasks, and address it in section 4. Mathematics Dataset Saxton et al. (2019) provides a broad range of this kind of task, such as algebra, comparison, polynomials, etc. Examples in the dataset include mathematical expressions and a small number of words in English.

**Binary Operation Table**  Power et al. (2022) presented the binary operation table. All examples in this task are expressed in the form $a \circ b = c$, where all five symbols are separately tokenized, and $\circ$ refers to arbitrary operations such as addition, subtraction, and some polynomial functions. In addition, the modulo $p$ is introduced, and the number of examples is finite.

**(Visual) Sudoku**  Sudoku is a puzzle game in which the player fills $9 \times 9$ cells with numbers from 0 to 9 without duplications in rows, columns, and $3 \times 3$ block areas. Visual Sudoku (Wang et al., 2019) is a special case in which images like MNIST represent numbers. Each symbol has fluctuations due to handwritten nature. We consider this game a combination of image recognition and a logical task.

**Abstract Reasoning Corpus (ARC)**  Abstract Reasoning Corpus (ARC) is proposed in Chollet (2019) as a benchmark for evaluating general intelligence skills. The task's input and output are a *grid* with ten colors. The input is several pairs of *grids* whose patterns can be recognized by humans using only a few examples. The problem setup represents few-shot supervised learning. The system should learn from a few examples, comprehend the pattern, and generate the grid following the pattern. The task is similar to the Raven's Progressive Matrices (Raven, 1936) but requires greater adaptation to new situations.

**SCAN**  SCAN (Lake & Baroni, 2018) is a supervised learning task in which models must learn to map commands to sequences of actions. It was designed to assess compositionality, which is the ability to understand or generate new combinations composed of known elements. In SCAN, the task is to transform commands into sequences of symbols of actions, but they are not tied to actual actions. gSCAN (Ruis et al., 2020) introduces a two-dimensional grid world, grounds the action sequences to the environment, and evaluates the compositionality.

**Grid World**  A grid world is an environment in which objects representing the agent, other items, and obstacles are placed on a two-dimensional grid. It often appears as a toy model in reinforcement learning. The agent must reach the goal through several procedures. Minecraft (Guss et al., 2019) can be considered a 3D version of Grid world, but because of its richer representation, it can be considered a combination of image recognition and a logical task.

## 4 Arithmetic Task

### 4.1 Arithmetic Task

Arithmetic tasks involve performing certain number operations to derive answers, such as addition, subtraction, etc. In machine learning, arithmetic tasks are often thought of as supervised learning in which a finite number of equations and their answers are given in the training phase. We can also introduce supervised

Table 2: **Accuracy of typical neural networks.** The values refer to MEAN(SD) of exact match (EM) in ten trials.

| Model | Parameter | Validation EM (%) | Test EM (%) |
|---|---|---|---|
| MLP | 1.3M | 6.91 (0.48) | 0.97 (0.08) |
| Seq2seq | 3.3M | **99.92** (0.20) | 1.34 (0.64) |
| Transformer | 3.2M | 85.99 (5.63) | **13.45** (3.26) |

signals by preparing the calculation process (Nye et al., 2022). In addition, these tasks can be treated as a task in reinforcement learning by introducing an environment and defining the basic operations as actions (Chen et al., 2018). In any case, the goal is not to achieve any abstract goal, such as acquisition of the concept of numbers, but rather to achieve arithmetic as an operation only.

There are two possible formulations for arithmetic tasks in machine learning: classification tasks, in which symbolic data are dealt with (Chen et al., 2018) (Nogueira et al., 2021) (Kaiser & Sutskever, 2016); and regression tasks, in which numerical data are used (Trask et al., 2018) (Madsen & Johansen, 2020). In this paper, we focus on symbolic representation. The simplest form would be a representation consisting only of mathematical expressions, as we do in the present study, but it is also possible to introduce natural language (Saxton et al., 2019) (Cobbe et al., 2021). It should be noted that the introduction of natural language makes the analysis and evaluation of these systems more complicated.

### 4.2 Addition Task

**General Problem Setup**  The addition task is considered a concrete example of the arithmetic task. For simplicity, consider the addition of two non-negative integers in decimal representation. The independent symbols are twelve symbols (i.e. 0, 1, 2, ..., 9, +, =). Given only a finite number of equations consisting of combinations of these symbols as training data, the task requires that operations be performed with high accuracy on the test data, which are unknown equations. This is often called an extrapolation regime when the equations are for numbers of larger digits that are not in the training data. Our usage differs from this definition; see Figure 1.

**Regarding Addition as a Logical Task**  We describe the addition task as a logical task by following section 3.1. Multiple symbols represent equations and answers, but what meanings are assigned to them is arbitrary. The addition task has only a few rules to remember, e.g., addition between two symbols and carrying. Under these rules, the result of an equation is entirely and uniquely determined. The rules have no exceptions and apply equally to all symbols. The answers are derived by repeatedly applying these finite rules to the parts. Solvers have to find and execute the algorithm, the process of deriving answers.

**Motives from a Logical Task Perspective**  An original purpose of Mathematics Dataset is to design new architectures. How neural networks which deal with logical operations to learn from data should be designed remains an open question, even for the addition task. There are many studies on Mathematics Dataset, as introduced in section 2.1, but the question has not been resolved yet from our point of view. Many studies solve the extrapolation issue by introducing task-specific inductive biases (Hu & Yu, 2020) (Nogueira et al., 2021) (Nye et al., 2022), or approaches such as LLMs (Henighan et al., 2020) (Chowdhery et al., 2022); those which do not introduce specific inductive biases have not achieved highly accurate extrapolation performance. This point is reproduced in section 4.3.2.

### 4.3 Experiment

We studied the extrapolation ability of neural networks in the addition task. Although this is the simplest arithmetic task, it is still sufficient to grasp extrapolation ability. In addition, the simple setup of this task has the advantage of allowing the behavior of what happened to be exactly and easily examined. We used pre-trained GPT-NeoX (20B) (Black et al., 2022) as an LLM, and MLP, Seq2seq (Sutskever et al., 2014)

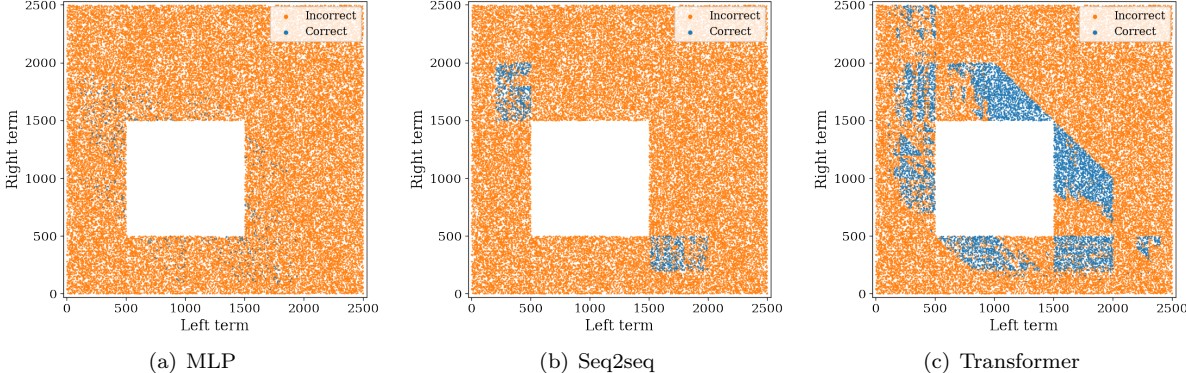

(a) MLP        (b) Seq2seq        (c) Transformer

Figure 1: **Extrapolation behavior on the small-digit dataset.** The figures show how neural networks succeed in prediction in the extrapolation regime. The axes refer to the values of the first and second terms in the equation. Training data are generated in $[500, 1500]$, which corresponds to the blank in the figures, and the test data - dots in the figures - are generated in $[0, 2500]$ as an extrapolation regime. Blue dots represent correct predictions (exact match), and the orange dots are incorrect.

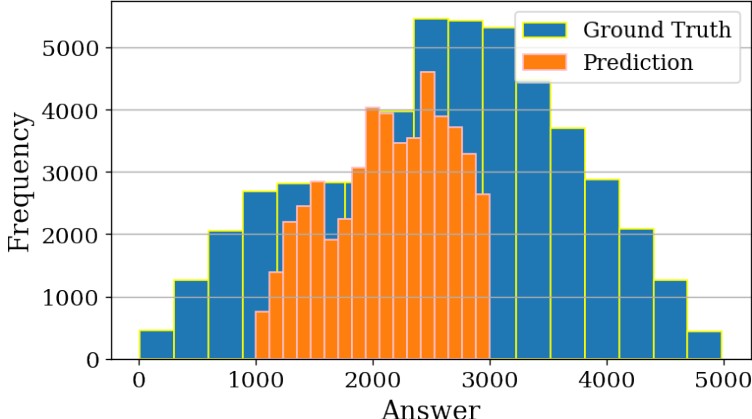

Figure 2: **Distribution of answers of Transformer.** The figure shows the distributions of the answer of Transformer to the ground truths (blue) and predictions (orange).

and Transformer (Vaswani et al., 2017) to study the behavior of representative neural network architectures. We trained all of these from scratch, except for GPT-NeoX.

The task is two-term addition for non-negative integers in decimal representation. The general problem setup of the addition task is described in section 4.2. The input is $a + b =$, and the output is $c$, which equals the result of calculation of $a + b$. $a, b$ and $c$ refer to integers and are represented by multiple symbols, e.g., $0, 1, \ldots, 9$, respectively. We deal with these symbols as independent tokens. One dataset for GPT-NeoX consists of large integers of up to 100 digits to investigate the response to large numbers and another of up to 4 digits to study the other neural networks. We call these the large-digit dataset and small-digit dataset, respectively. The details of the experimental setup are shown in Appendix A.

### 4.3.1 Extrapolation Behavior of Typical Neural Networks

We trained MLP, Seq2seq, and Transformer on the small-digit dataset. Table 2 shows the accuracy of these models. MLP cannot generalize even in the interpolation regime, while Seq2seq can generalize well in the interpolation regime but not in the extrapolation regime. Generalization by Transformer is worse than that by Seq2seq in the interpolation regime, but gives the best accuracy in the extrapolation regime. Our result is consistent with Saxton et al. (2019).

Table 3: **Results for prediction with GPT-NeoX in the large-digit dataset.**

| Group | Number of examples |
|---|---|
| Incorrect with non-numerical tokens | 66,531 |
| Incorrect with numerical tokens | 32,819 |
| Correct | 650 |
| Total | 100,000 |

The training data is inside the square $500 \leq a, b \leq 1500$, which corresponds to the blank in Figure 1; we call this an interpolation regime. Figure 1 shows that Transformer successfully acquires the extrapolation ability to a certain extent, whereas MLP and Seq2seq do not. Figure 1(c) shows that Transformer succeeds in extrapolating, although the extrapolation regime is limited. Interestingly, the area is bounded in $1000 \leq a + b (= c) \leq 3000$ (Figure 2). The lower (upper) bound corresponds to the training data's minimum (maximum) value. This would be because predicting numbers that do not belong to the range does not contribute to minimizing loss.

### 4.3.2 Extrapolation Behavior of an LLM

We experimented with pre-trained GPT-NeoX (20B) to study how an LLM behaves with large numbers. We generated 100,000 prompts and obtained a prediction for them. The prompt is "What is $a + b$?", where $a$ and $b$ refer to numbers up to 100 digits. We clarified the responses into three groups: incorrect with non-numerical tokens, incorrect with only numerical tokens, and correct, and counted the number of examples in each group (Table 3). The number of correct answers is 650.

Figure 3 shows that error increases and becomes larger as the number of digits increases. Figure 4(b) shows the tendency for predictions to remain in the range of $[10^{10}, 10^{45}]$ even though the ground truths are large.

## 5 Discussion

### 5.1 Our Experiment

**Extrapolation Behaviors**  How generalization occurs depends on the architecture. This suggests that Transformer is more suitable for this task than the others. However, as we can see from the standard deviation of the test EM, this behavior does not seem to be stable (see appendix B.2 in detail). The area that the Transformer successfully predicts changes depending on the value of the random seed.

Nogueira et al. (2021) found that T5 (Raffel et al., 2020) generalizes to larger digits after introducing position token as a representation of numbers. Experiments to study extrapolation behavior with such a representation are worthwhile, as the results will reveal how the data representation contributes to the extrapolation.

Our experiment and the related work indicate that finding a suitable data representation, adding more data related to the target task, and introducing elaborated architectures such as Transformer are good inductive biases. However, relying on these approaches only cannot control the behavior of models in an extrapolation regime, and systematic behavior cannot be expected.

**Typical Errors**  Figure 4(a) shows that dots are distributed along certain lines. $y = x$ means that the prediction matches the ground truth. Many dots are distributed along the line $y = x + 1000\,\beta_1 + 100\,\beta_2 + 10\,\beta_3 + \beta_4$ where $\beta_i = -1, 0, 1$. This means that the prediction was wrong, with several carrying errors. We also see some stripes for $y = 10x + \beta$, which means that the model gives incorrect $\langle EOS \rangle$ at the right-most digit.

**Effect of the Training Dataset**  GPT-NeoX was trained on the dataset Pile (Gao et al., 2020), which includes many text data in different contexts as well as Mathematics Dataset and arXiv preprints to com-

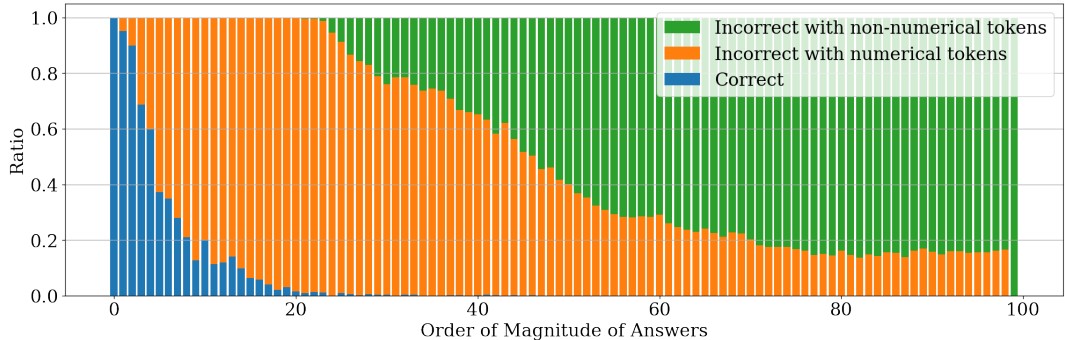

Figure 3: **Ratio of correct and incorrect prediction of GPT-NeoX.** The figure shows the ratio of correct predictions (blue), incorrect predictions due to the inclusion of non-numerical tokens (green), and incorrect predictions even using only numerical tokens (orange).

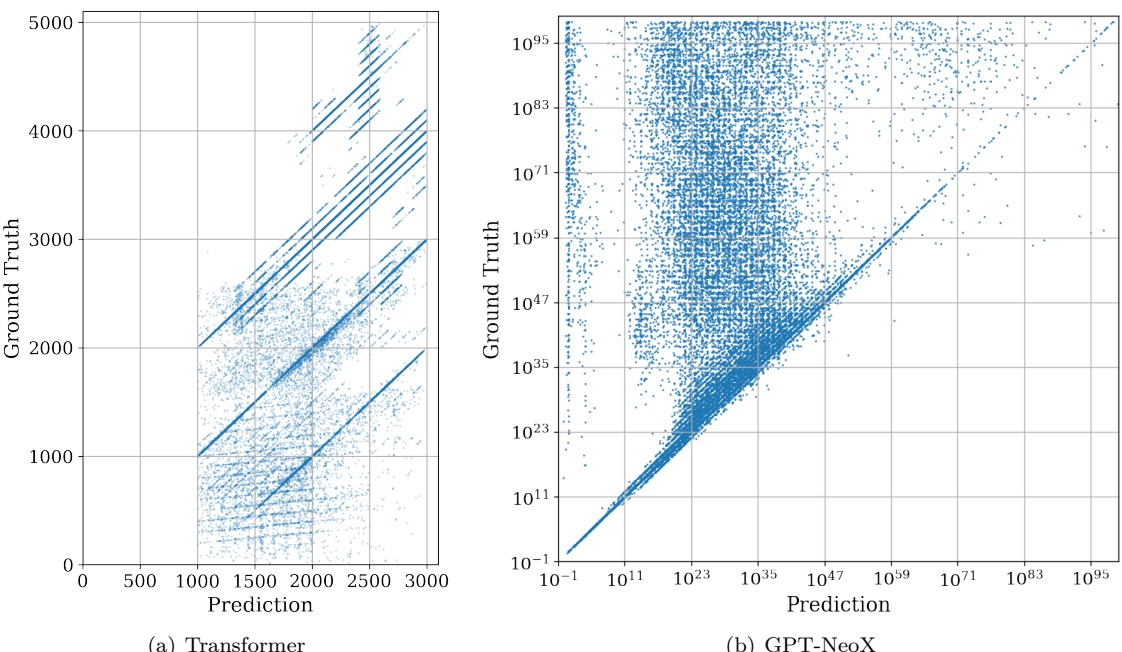

(a) Transformer

(b) GPT-NeoX

Figure 4: **Prediction vs. Ground truth.** The figures show how the prediction differs from the ground truth. The $x$-axis refers to the prediction, and the $y$-axis to the ground truth. (b): Dots refer to both "incorrect with numerical tokens", and "correct" in Table 3.

pensate for the mathematical reasoning. Answers in the *add_or_sub* and *add_or_sub_big* modules in Mathematics Dataset vary by digit number, and range up to 20 digits (see Figure 5 in Appendix B). As Razeghi et al. (2022) pointed out, the kind of data used affects the competency of the model. The distribution range coincides with the fact that the error becomes large from around 20 digits (Figure 4(b)).

In principle, the same phenomenon occurs no matter to what degree the training data are increased. No matter how large the digits used for training, neural networks cannot behave like humans. In other words, humans can calculate any digits without training on huge numbers, but neural networks cannot, and the desired behavior is not completely ensured.

## 5.2 Logical Task and Extrapolation

Extrapolation is the central issue in machine learning, but at the same time, it is difficult to discuss because of the lack of a clear definition. In a supervised learning setup, it might be impossible to make correct inferences from the test data by training on finite training data. In principle, the training examples do not have to have information about the test examples. For instance, a finite set of equations about integers in the addition task cannot uniquely determine the relationship for all integers.

The logical task forces us to solve this kind of difficulty. The logical task is strongly related to extrapolation, which can be a clue to tackling this challenge. Humans can add any two large numbers, albeit allowing that lengthy solution procedures may be attended with error. Extrapolation by humans also depends on logical operation. We can do this because we grasp the rules behind the dataset and operate some operations based on the rules. We do not just directly memorize all examples we have seen before. Mere memorization does not function when new examples are encountered.

The open question is how do humans learn these rules and accomplish this task. We do not simply rely on the pairs of inputs and outputs, but rather also on the description of the task, usually described in natural language, gesture, some instruction, and so on. This kind of instruction is inevitable for a system to learn complicated tasks, but might not always be necessary, especially for easy tasks. As a first step, it is better to focus on simple tasks as presented in section 3.4 to find essential inductive biases.

In Power et al. (2022), Transformer seems to succeed in having the complete algorithm in binary operation table tasks, albeit that the dataset size is finite. This and our result suggest that Transformer, or the attention mechanism, is a good starting point to tackle the logical task.

## 5.3 Inductive Bias for Logical Task

Building systems that solve logical tasks without any inductive bias is impossible. In arithmetic tasks, successful prediction in the extrapolation regime is a matter of concern. Even in simple addition tasks, there are infinite examples (i.e., numbers with large digits), and it is not easy to make correct predictions. Below, we discuss inductive bias, which we expect to be widely influential in arithmetic, or more general logical tasks.

**Modularity of Neural Networks**  Since logical tasks are composed of finite rules and solved with operators, the use of neural networks with modules seems a promising approach. Kirsch et al. (2018) proposes the Modular Network, which consists of multiple modules contained within it; the same module is used for similar examples to reduce the number of parameters. However, it is not clear what functions the introduced modules possess. Csordás et al. (2021) introduces a method to study whether functional modularity in neural networks spontaneously occurs and observes that the reusability of weight for the same function is not endowed, and that specialization is not particularly strong. With further development, these studies may allow the training of neural networks with functional and rule unit modules.

**Explicit Division of Processing**  Neural networks that make predictions through explicit stepwise processing to accommodate the multi-step nature of logical tasks can be considered. Banino et al. (2021), Schwarzschild et al. (2021) and Bansal et al. (2022) propose recurrent neural networks that perform multiple recursive processes depending on the complexity of the task. The state of recursive processing is maintained as the state of the recurrent neural network. We might extend this to output explicit discrete representations as an intermediate state toward the final answer. In other words, solving by parts and accumulating small steps that can be confidently determined to be correct yield the final answer. For example, the intermediate results of operations for each digit represent an intermediate state in addition or multiplication. Another example is filling cells partially in sudoku.

Furthermore, in this process, the system solves the task in parts, and the same operations appear repeatedly in different places. Hence, finding finite fundamental operations for the task is helpful in securing the stability of predictions, and obviates the domain-specific. On the other hand, the modularity of the neural networks

does not appear naturally (Csordás et al., 2021); an inductive bias is still required, such that neural networks have modules corresponding to operations in the logical tasks.

**Task Description**   In the present machine learning paradigm, pairs of input and output are given in the supervised learning setting. This format forces us to handle only a finite number of data points. When humans do equivalent tasks, we learn from the finite dataset and the description that describes the task's details in natural language, gestures, etc. If this problem setup difference interrupts successful prediction in extrapolation regimes, it can be improved. This can be termed *task description*, and be provided in addition to the dataset.

Task description is a further representation of the task which differs from the input-output pairs. On the one hand, the question is whether it is possible to build a system that follows the task description and solves the task. On the other hand, it can be seen as a representation that covers out-of-distribution data points in advance. We can then view the issue as finding a function that connects the inputs and outputs with another broader, more general representation that refers to the relationship between them. We may then consider all examples outside the training data as the interpolation regime, given that they are accommodated in the task description.

At most, the number of task descriptions given for a task is small. Task descriptions are therefore not usable in machine learning, which usually requires many data points. However, if the recent LLMs are beneficial in grasping the meaning of texts, these pre-trained models might be applied to analyze the description: they might work in constructing and executing algorithms to solve logical tasks.[2]

### 5.4   Logical Task and Explainability

As mentioned in the previous section, algorithms or multi-step processes can solve logical tasks. Solving through multi-step operations can be seen as decomposing the task into subtasks. The decomposition of the task is a means of explaining the task, and makes it easy to identify where the system fails. If a system solves a logical task in this way, the explainability of the system is higher than that of neural networks which lack any inductive bias for logical tasks. On the other hand, this multi-step process is not necessarily observed outside the system. In this case, it does not contribute to the high explainability, but depending on whether or not the system's action is based on the way the task is decomposed, the system affects the accuracy of the extrapolation regime.

Moreover, if the operations are restricted to a finite set and the system uses them repeatedly, the system's behavior is bounded and gives interpretability. This situation is similar to the reinforcement learning setup, wherein the agent has to choose a sequence of pre-defined actions, and the range of the behavior becomes predictable. On the other hand, although the behavior is bounded, it provides no information on why the agent chose the action; understanding the policy is thus a different issue.

To solve a logical task is to execute an algorithm to solve the task. A system that can faithfully follow an algorithm provides reliability. It is nontrivial that a neural network could be such a system. However, consistent with the concept of connectionism, it would be possible to create a system that is as faithful to the algorithm as are humans. Moreover, once a valuable approach to the logical task is found, it may reduce the chance of making mistakes even in fields separate from pure logical tasks.

## 6   Conclusion

This paper aims to integrate logical processing with the inference capabilities of recent deep learning, a goal which has long been sought. To this end, we first defined logical tasks. A logical task consists of three elements:

- Symbols representing the input-output pairs.

---

[2]Mishra et al. (2022) provides the dataset with instruction annotation, which can be considered as a kind of the task description.

- A finite number of rules between symbols.

- Existence of an algorithm to solve the task.

The actual content of logical tasks is a collection of previously proposed tasks. Here, however, by looking at the process from a higher perspective and capturing common properties, we aimed to develop a broad and effective inductive bias that is not restricted to individual tasks. Our perspective provides the foundation for developing inductive biases that are not specific to a single task but rather have broader impact, such as on convolutional neural networks for image recognition and Transformers for serial data.

The definition of logical tasks allows us to discuss the system requirements for solving them. Furthermore, we have provided more specific directions for developing several possible inductive biases. Their development is of the utmost importance to future work. We also noted that it is not critical to solve individual tasks simply but to consider developer knowledge and not to be too task-specific. This can either be mitigated by conscious developers or by having a variety of tasks, as we have presented. Although beyond the scope of this paper, implementation of usable tasks and baseline models are the next crucial steps in this work.

The rationale for the defining of logical tasks in the first place lies in the lack of inference capability of neural networks for arithmetic tasks. Although this is a well-known fact, we also reproduced that their extrapolation fails in a simple experimental setting and with new visualizations. Our experiment setup and visualization offer a new direction for examining the extrapolation ability of neural networks.

Our direction differs from attempts to find explanatory possibilities by analyzing trained neural networks, as in XAI. We are aiming at systems that, by their behavior, perform logical processing, i.e., behavior that representable by highly explainable, white-box algorithms. Other properties we hope AI systems will possess include generalization and learning from small amounts of data. By discussing the relevance of these concepts to logical tasks, we have clarified the significance of solving them. Since NeSy AI's motivation is close to ours, our discussion can be seen as a discussion of the desired benchmark direction in this area.

Extrapolation is a central and essential issue in machine learning but is often used as an ambiguous term. Addressing logical tasks, i.e., aiming to make coherent predictions for an infinite number of data points, is therefore very closely related to the issue of extrapolation. Even if the data points are infinite, the rules are finite, and capturing them should be critical to building a system that systematically succeeds or fails at extrapolation. This view cannot be obtained if taken as mere curve fitting.

We hope that the logical task and perspective we propose will lead to the invention of innovative inductive biases.

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

## A   Experiment Setup

We performed two types of experiments. The first aimed to study the extrapolation behavior of neural networks with typical architectures such as MLP, Seq2seq, and Transformer, and the second used GPT-NeoX(20B) as an LLM. As in natural language processing, numbers are represented by multiple tokens. We used two different datasets for the addition task.

### A.1   MLP, Seq2seq and Transformer

#### A.1.1   Dataset

We generate 40,000 and 5,000 pairs of random integers without duplication in the $[500, 1500]$ range for training and validation. We generate 50,000 pairs for the test data in the range $[0, 2500]$, excluding those in the training range. In figure 1, the blank area corresponds to the range of the training data, and dots refer to examples in the test data. All data are sampled from the uniform distribution. We call this dataset the small-digit dataset in the paper.

### A.1.2 Common Hyperparameters

The following hyperparameters are used for every architecture. The batch size is 256. The loss function is the categorical cross-entropy loss. The number of epochs is 100, enough to saturate the accuracy for all models. We adopt the best validation EM model to calculate the test EM in each trial.

### A.1.3 MLP

The model consists of four fully-connected layers with 512 hidden units, dropout layers, ReLU as the activation functions, and Softmax as the output layer. The optimizer is Adam (Kingma & Ba, 2015) with the learning rate of 0.001, $\beta_1 = 0.9, \beta_2 = 0.999$ and $\epsilon = 10^{-8}$.

### A.1.4 Seq2seq

Seq2seq is composed of an encoder and decoder with recurrent units. These comprise the embedding layer with 512 dimensions for embedding and GRU (Cho et al., 2014) as the recurrent unit with 512 hidden units. The optimizer is Adam with the learning rate of 0.001, $\beta_1 = 0.9, \beta_2 = 0.999$ and $\epsilon = 10^{-8}$.

### A.1.5 Transformer

The Transformer we used is the encoder-decoder type. The number of layers for both the encoder and decoder is 3. The number of multi-head attention is 8. The embedding dimension is 256. The dimension of the feedforward network is 256. The optimizer is Adam with the learning rate of 0.0001, $\beta_1 = 0.9, \beta_2 = 0.98$ and $\epsilon = 10^{-9}$.

## A.2 GPT-NeoX(20B)

### A.2.1 Dataset

The setup is the same as NLP. Multiple tokens represent numbers. The prompt form is "What is $a + b$?" where $a$ and $b$ are integers. We generated 1-digit to 100-digit integers. We generated 200,000 integers, that is, 100,000 pairs, as the number of digits follows a uniform distribution. We call this dataset the large-digit dataset in the paper.

### A.2.2 Parameters

We used pre-trained GPT-NeoX(20B) with maximum_tokens of 105, temperature of 0.1, top_p of 0.0, top_k of 0, recompute of false, and number of samples of 10. We used default values for other parameters.

# B Supplementary Experiment Result

## B.1 GPT-NeoX(20B)

Figure 5 shows the distribution of answers in Mathematics Dataset (Saxton et al., 2019), which was used for pre-training of GPT-NeoX(20B). Figure 6 shows how the correct and incorrect data points are distributed. While the behavior is almost symmetric with respect to swapping the first and second terms in the experiment of the small-digit dataset, it is asymmetric in the large-digit dataset. Given that the prompt is provided as a one-dimensional series, this asymmetrical behavior is likely due to large $N$-digit numbers.

## B.2 Analysis of Transformer Trained on the Small-digit Dataset

**Dependencies on Random Seed**   In this experiment, we ran ten trials for every architecture. Figure 7 shows how Transformer succeeds in predicting the extrapolation regime. It shows the dependency of random seeds and the area of blue dots shows distributing changes. This result is consistent with Nogueira et al. (2021) as they also claimed that the behavior in the extrapolation regime is unstable.

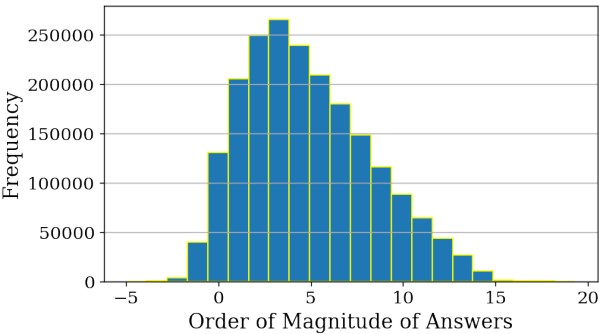

Figure 5: **Distribution of answer in Mathematics Dataset.** The figure shows the distribution of the answer in modules *add_or_sub* and *add_or_sub_big* in Mathematics Dataset.

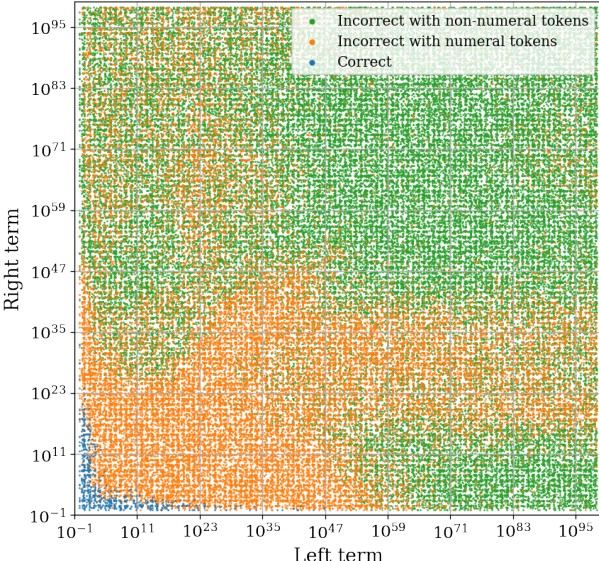

Figure 6: **Extrapolation behavior of GPT-NeoX on large-digit dataset.** The figure shows how GPT-NeoX succeeds in prediction in the extrapolation regime. The axes are the values of the left and right terms of the equation. Blue dots represent correct predictions (Exact Match), orange represents incorrect with only numerical tokens, and green represents incorrect with non-numerical tokens.

**Top-100 Errors** Figure 8 shows how many errors are found in a trial. Many errors include the symbol 1. Errors of 1000, 0, and -1000 account for nearly 50% of the total. This means that carrying errors are dominant.

## B.3 Transformer Trained on the Larger Small-digit Dataset

**Extrapolation Behavior** The extrapolation behavior changes as in Figure 9. The behavior on the extrapolation regime close to the training data is stable. We can also see unstable areas in the upper right in the figures. The area appears above the line $y = 5000 - x$. However, since the extrapolation behavior is not stable and mostly gives wrong predictions, our conclusion that we need novel inductive biases remains unchanged.

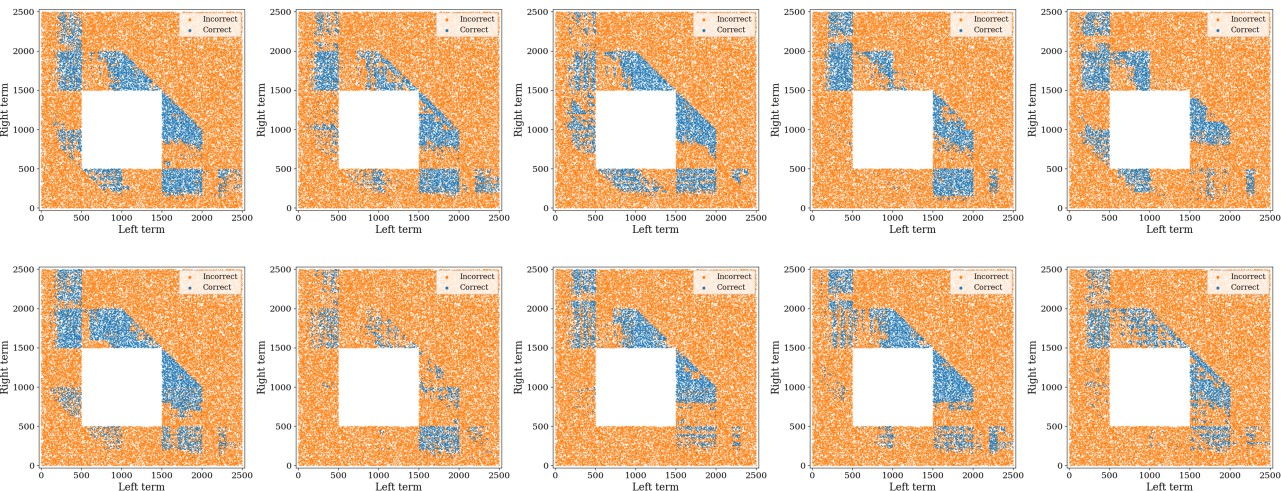

Figure 7: **Random seed dependency of the Transformer.** We ran ten training procedures with different random seeds. We use the best validation epoch for every model to plot the figures. The area that the model succeeds in predicting is not stable.

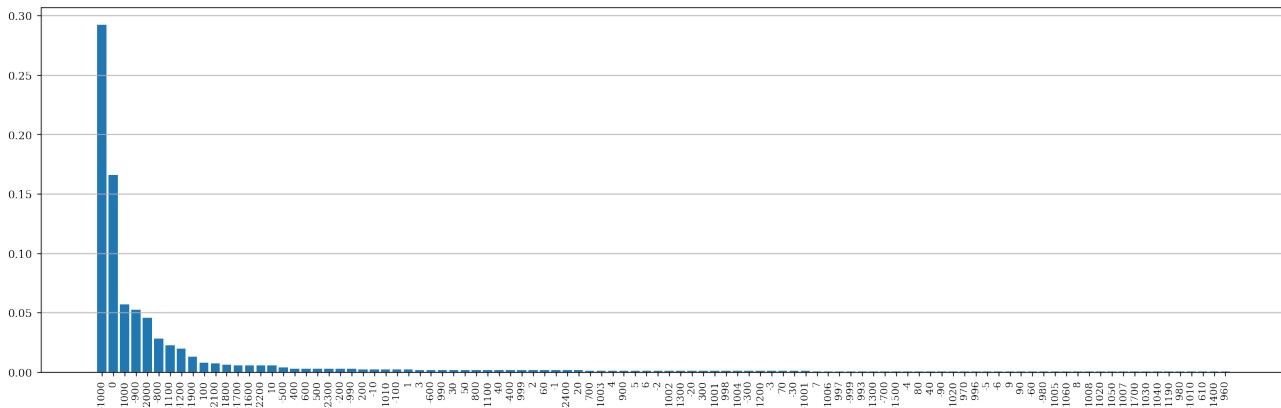

Figure 8: **Top 100 of Errors of Transformer.** The figure shows the top 100 errors in a trial.

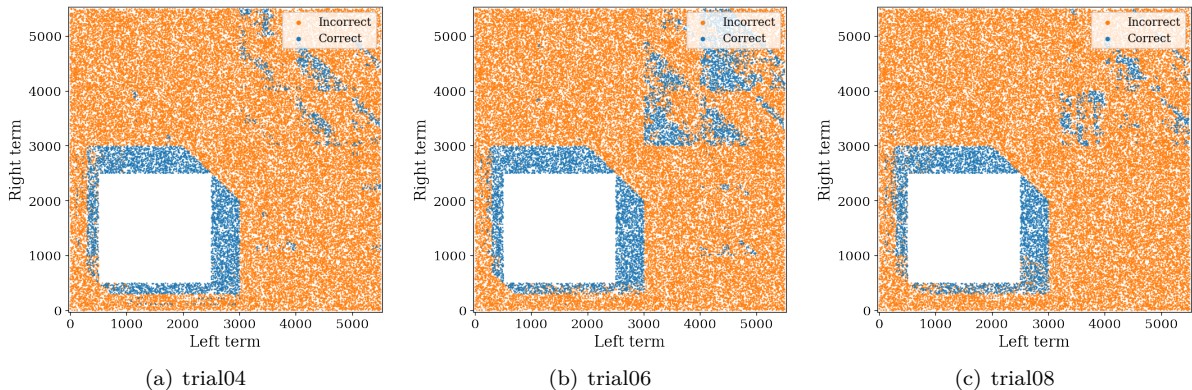

(a) trial04  (b) trial06  (c) trial08

Figure 9: **Extrapolation behavior for the larger small-digit dataset.** The figures show how neural networks succeed in prediction in the extrapolation regime. The axes refer to the values of the first and second terms in the equation. Training data is generated in $[500, 2500]$, which corresponds to the blank in the figures, and the test data - dots in the figures - are generated in $[0, 5500]$ as an extrapolation regime. Blue dots represent correct predictions (exact match) and orange are incorrect.

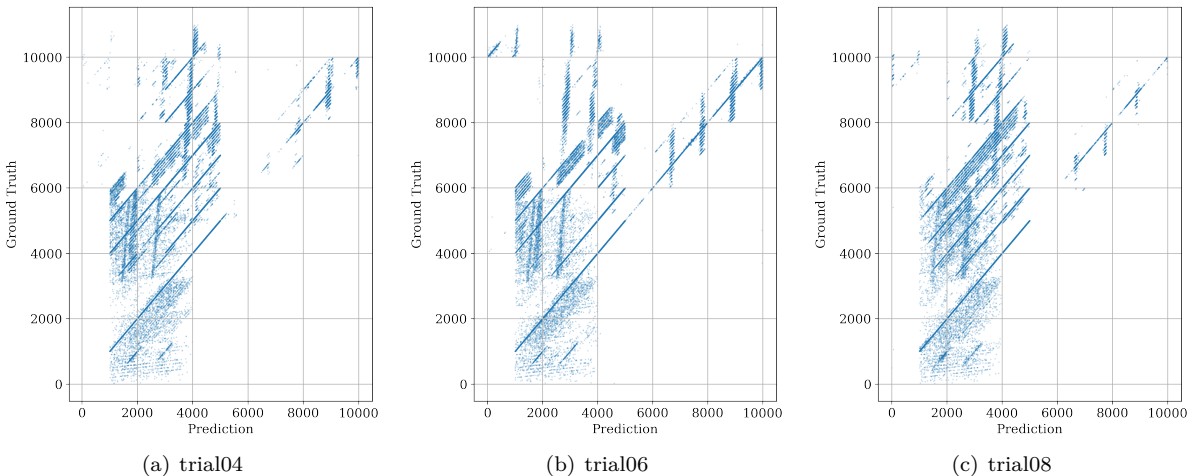

(a) trial04          (b) trial06          (c) trial08

Figure 10: **Prediction vs. Ground truth. for the larger small-digit dataset.** The figures show how the prediction differs from the ground truth. The $x$-axis refers to the prediction and the $y$-axis to the ground truth.

