# OpenReview forum: "Logical Tasks for Measuring Extrapolation and Rule Comprehension"
_TMLR — Withdrawn by Authors_

### Review · Reviewer_dSsa · 2022-12-20

**Summary Of Contributions:**

The main contribution of the paper is methodological and the paper should be evaluated based on this. It includes the value of the proposed "logical tasks" (at overcoming the limitations of current AI systems), the "experimental setup" introduced in the paper to study extrapolation, and the negative results obtained "networks cannot extrapolate". Next, I consider these in turn.

(1) logical tasks: in the attempt to define "symbol" the point made about "fluctuations of representations" is very unclear to me. The definition of rules is equally vague. Are these referring to Peano arithmetic? The confusion increases when "system requirements" are discussed. Section 3 would require a full rewrite. Surprisingly, the "definitions" are informal. The only useful part of Section 3 are the examples at the end, which describe the tasks/data sets used, but here too there are missing references. This is a selection of "logical" tasks, but they don't live up to the claim in the Introduction: very little is said as to why these choices were made and how they might relate to each other to form a representative collection of logical tasks.

(2) experimental setup: the authors limit their analysis to language models, and limit as a result their survey of NeSy approaches to Hamilton et al. 2022, presumably because of the choice to focus on language models, but since the main task at hand is reasoning there is quite a gap in the NeSy literature to be closed in the paper. Having said that, the arithmetic task that is considered in the paper is based on symbolic representation without natural language. The architecture that is used at this point is unspecified and with this the results in Table 2 are difficult to appreciate. Figure 1 is useful but the "extrapolation regime" does not seem to have been specified. Perhaps the paper would be better received if it had focused with more detail on the definition and evaluation of such extrapolation regime rather than discussing other broader claims and general questions of reasoning.

(3) extrapolation results: The authors point out the imprecise nature of the use of the term extrapolation in relation to generation and in particular OOD generalization. However, I was hoping to see as a contribution of this paper a formal definition of extrapolation. Perhaps this should be provided with a caveat that the definition might not apply outside of math problems, but in this area it should be possible to provide bounds on what is meant by extrapolation, maybe by borrowing from the definitions of robustness from the area of neural network verification.

**Audience:**

Yes

**Claims And Evidence:**

No

**Requested Changes:**

Please see above

**Strengths And Weaknesses:**

When it is said that "PaLM (Chowdhery et al., 2022), shows tremendous ability to provide plausible outputs in reasoning
tasks, such as explaining jokes and chaining inferences" I'd recommend qualifying the uses of the word "reasoning" in the paper. Explanation-based reasoning is quite different conceptually from the inferences of commonsense reasoning used by humans to make sense of a joke. The main topic of the paper, deductive and mathematical reasoning, is also very different from human-like reasoning.

While I agree with the main arguments put forward in the paper (data efficiency, extrapolation), I am not surprised by the conclusion that the model's mathematical reasoning is limited. Why should the mathematical reasoning capacity of such models not be highly limited? Is there anything in the architecture or the way that the model is trained that might point to reasoning being a capability of the system? Contrast with the DeepProbLog approach (https://arxiv.org/abs/1805.10872) where the primitives for arithmetic computation are provided symbolically. Other neurosymbolic approaches could have been considered (see https://arxiv.org/abs/2012.05876 for an overview).

I do value the use of arithmetic in the evaluation of large neural network models but from the perspective that such study may help shed light, in a precise way, into the main ingredients of such large models and their limitations. I am less convinced that such large models should be expected to be useful for mathematical reasoning without the help of a symbolic layer. It seems to me that, at a minimum, the key concepts of consistency and coherence (see https://discovery.ucl.ac.uk/id/eprint/10155813/) should be instilled into the model before one can have any hope of producing effective results.

The authors note that "overcoming this situation requires the development of novel architectures or inductive biases. However,
inventing these is no easy task. As a prior step, in this paper we review the current situation, clarify the
problem, and indicate directions". This is fine, but the paper seems to ignore very relevant work in neurosymbolic AI which already started to answer the question of architecture and inductive bias, although these have not been applied directly to large scale studies.

---

### Review · Reviewer_XBAN · 2023-01-13

**Summary Of Contributions:**

This paper makes two main contributions:

**First**, it defines "logical tasks" as a key challenge for current AI systems, especially large language models, and unifies several existing logical reasoning/mathematic/algorithmic tasks under this framework.

In the author's words, logical tasks are defined by input and output symbols, where the output is determined by some latent but ideally deterministic algorithm. Examples include multi-digit addition, systematic instruction following benchmarks like SCAN (Lake and Baroni), the Abstract Reasoning Corpus (ARC) of Chollet (2019), etc.

The authors explain that some key challenges faced by models trying to solve logical tasks are
1. an inability to rely on pretraining data (as LLMs like GPT-3 or PaLM do), since symbols in a logical task be artificial or arbitrary (e.g. ARC)
2. extrapolation from the training distribution to problems outside the training distribution (e.g. generalization from 2-digit addition to 3-digit addition)

**Second**, the paper provides an investigation on an example logical task: that of multi-digit addition. The authors evaluate various model architectures, including MLP, seq2seq (RNN-based) model, and a large language model, GPT-NeoX, on the ability to learn addition problems both in- and out-of-domain. The authors claim that this illustrates the suitability of transformers for solving logical tasks, but note that there are several confounds for this experiment (see Weaknesses).

**Audience:**

Yes

**Broader Impact Concerns:**

None that I can see. Nice review of the SoTA and the literature.

**Claims And Evidence:**

No

**Requested Changes:**

In order to secure my acceptance recommendation, I would expect (1) an experiment without confounding factors, so we can more clearly buy the authors' central claim that Transformer architectures generalize better on multi-digit addition; and (2) a set of results beyond addition that draw some conclusions about the group of "logical tasks" defined by the authors, since addition has been a very frequently studied task in the literature thus far.

**Strengths And Weaknesses:**

# Strengths

- The paper correctly identifies and reviews a crucial weakness for AI systems, which the authors label "logical tasks". Authors are correct that despite increasing pretraining data, large language models (and other foundation models) still struggle at tasks involving logical and algorithmic reasoning.
- There are a number of interesting visualizations of extrapolation behavior for transformers in the experiments section.

# Weaknesses

I see two main weaknesses with this paper:

**First**, it is not clear how useful of a contribution this "logical task" framework is. The review of the existing literature is nice, but it's not clear that unifying constitutes a substantive contribution to the literature. I believe most ML researchers, including TMLR's audience, probably already has some intuitive category in their mind about "tasks requiring logical reasoning", under which additon, SCAN, ARC, etc all fall. It's nice to make it explicit here, but this literature review, by itself, does not warrant publication at TMLR.

**Second**, the main experiment in the paper: the evaluation of MLP vs seqseq vs Transformers on addition—is unfortunately technically unsound. **There are several confounds in this study that the authors do not address.** First, model size: there are no model size details given, but presumably the MLP and seq2seq models are much smaller than gpt-neox. So this experiment doesn't really tell us much, besides that lager models seem to do better on addition tasks. Second, pretraining: GPT-NeoX is finetuned, not trained from scratch.

Thus, it's not clear what central message readers should get from the expeirment in this paper. We **certainly** cannot get the takeaway statement in the discussion that "How generalization occurs depends on the architecture...this suggests that Transformer is more suitable for this task than the others." Again, if authors want to make this claim, they need to control for as many confounding factors as possible.

Finally, even if these results were sound, their relationship to existing work needs to be made clearer. In section 4.3.1 authors note that their results are consistent with "Saxton et al. (2019)". What are the takeaways for readers beyond  just reading Saxton et al. (2019), then? Indeed there the authors cite a large literature on extrapolation of language models on "logical tasks". Most of these papers observe similar phenomena on the exact same task, addition, that is studied here (e.g. Nogueira et al., 2021, Nye et al., 2021). How do the authors' results add to this conversation?

# Conclusion

Overall, while I appreciate the authors' scholarship in unifying several tasks under this "logical task" umbrella, and thought there were several interesting visualizations in the paper, unfortunately I find that (1) the "logical task" framework is not substantive enough to warrant a full paper, and (2) the core experiment in the paper has several confounding factors, which makes me unable to recommend acceptance at this time.

# Minor

- The citations at the end of page 2 to me are suspect. It's not clear to me how Chain-of-Thought prompting is a specialized "module" tailored to the characteristics of the task. Simliarly, Nogueira 2021 does not introduce new modules, but rather just evaluates the ability of simple transformers to solve mathematical reasoning tasks.
- End of section 2.2: "Mathematics requires strict multi-step operations and is therefore an archetypal task for demonstrating the shortcomings of LLMs. There are no reliable large language models to do this with symbolic operations". It's a little unclear whether this is really a shortcoming. LLMs like GPT-3 are arguably not designed/trained with the sole purpose of being a reliable arithmetic calculator. If we finetuned GPT-3 continuously on endless mathematical datasets and equations, then surely we would see much greater performance. The inability of GPT-3/etc to solve math problems zero- or few-shot is not necessarily a limitation of a large language model in and of itself, but rather just limitations to what abilities can be learned from web-scale text data.
- Figure 1 is a cool plot, but it doesn't give us a clear understanding of a model's overall accuracy in various extrapolation regimes - it's obvious there are blue dots here and there, but the proportion of blue dots to orange dots is unclear. More clear would be a heatmap-style visualization that reports **percent accuracy** for various binned regions outside of the training dataset
- Table 2 "Test" and "Val" columns should be much clearer which are interpolation regime and which are extrapolation. I'm assuming Val is interpolation and Test is OOD but this is not clear.
- Sections 2.2, 2.3, 2.4, 2.5, 2.6 seem a little overly didactic, and probably unnecessary for a single paper (e.g. I don't think we need to reiterate the one-dimensional example of extrapolation vs interpolation).
- More detail for some of the neural architectures here are needed. How is an MLP used to add digits? What is the input/output format?

---

### Review · Reviewer_h2Z7 · 2023-04-17

**Summary Of Contributions:**

The authors propose a task domain, named logical task, as a new research direction for machine learning. They try to propose a definition of logical tasks and demonstrate an example from this domain, integer addition. They reported results of MLP, Seq2Seq, and Transformer, which are consistent with David Saxton, Edward Grefenstette, Felix Hill, and Pushmeet Kohli ICLR 2019. They also tried an LLM, GPT-NeoX. They conclude that their proposed task domain deserves more attention from the community.

**Audience:**

No

**Claims And Evidence:**

Yes

**Requested Changes:**

1. The authors may want to consider trying their dataset on GPT-4.
2. The authors may want to extend their "logical tasks" to regimes outside of integer addition
3. The authors may want to try different sizes of training data


**Strengths And Weaknesses:**

I would recommend a resubmission to some other venueIN since the current version does not meet the standard of a TMLR paper from my perspective.

In particular, here are some major weaknesses that the authors may want to pay attention to:

1. In the introduction, the claim that neural networks "do not have mechanisms suitable for utilizing seen data repeatedly, nor the ability to explicitly and efficiently use patterns and rules hidden in the dataset" is unconvincing. Actually, neural networks are pretty good at pattern recognition as universal function approximators.

2. In 2.2, the authors' claim that no large language model can do "symbolic operations" lacks evidential support. The LLM they experiment with is not the SOTA.

3. In 2.5, the authors discuss the up and down side of having inductive bias but do not provide a concrete way to evaluate how much inductive bias is not extreme. The methodological inquiry proposed here is vague and not very helpful.

4. In 2.6, the authors use the concept of out-of-distribution without defining it. "Since OOD generalization, from its definition, argues for prediction on data not seen in the training dataset, it can also be regarded as a kind of extrapolation" is an inappropriate claim since even for interpolation the testing data is not supposed to be seen in training sets.

5. In 3.1, the definition of logical tasks is "tasks in which symbols represent inputs and outputs, the symbols are subject to rules, and an algorithm utilizing the rules determines the outputs from the inputs unambiguously". It looks pretty vague to me. The same applies to the definition of rules. "Every logical task has its own set of rules. This set of rules determines the relationships among symbols" does not convey anything concrete. "We define white-box algorithms as those algorithms that are short enough that humans can represent them in natural language or programming language and execute them in practice." What do you mean by "short enough"? Is a recursion short enough?

6. The weakness of the experiment can be found in the next part.

---

### Note · Authors · 2023-05-08

I have read and agree with the venue's withdrawal policy on behalf of myself and my co-authors.